# Causal Imitation Learning
# with Unobserved Confounders

**Junzhe Zhang**
Columbia University
junzhez@cs.columbia.edu

**Daniel Kumor**
Purdue University
dkumor@purdue.edu

**Elias Bareinboim**
Columbia University
eb@cs.columbia.edu

## Abstract

One of the common ways children learn is by mimicking adults. Imitation learning focuses on learning policies with suitable performance from demonstrations generated by an expert, with an unspecified performance measure, and unobserved reward signal. Popular methods for imitation learning start by either directly mimicking the behavior policy of an expert (*behavior cloning*) or by learning a reward function that prioritizes observed expert trajectories (*inverse reinforcement learning*). However, these methods rely on the assumption that covariates used by the expert to determine her/his actions are fully observed. In this paper, we relax this assumption and study imitation learning when sensory inputs of the learner and the expert differ. First, we provide a non-parametric, graphical criterion that is complete (both necessary and sufficient) for determining the feasibility of imitation from the combinations of demonstration data and qualitative assumptions about the underlying environment, represented in the form of a causal model. We then show that when such a criterion does not hold, imitation could still be feasible by exploiting quantitative knowledge of the expert trajectories. Finally, we develop an efficient procedure for learning the imitating policy from experts' trajectories.

## 1 Introduction

A unifying theme of Artificial Intelligence is to learn a policy from observations in an unknown environment such that a suitable level of performance is achieved [33, Ch. 1.1]. Operationally, a policy is a decision rule that determines an action based on a certain set of covariates; observations are possibly generated by a human demonstrator following a different *behavior policy*. The task of evaluating policies from a combination of observational data and assumptions about the underlying environment has been studied in the literature of causal inference [29] and reinforcement learning [37]. Several criteria, algorithms, and estimation methods have been developed to solve this problem [29, 36, 3, 6, 35, 32, 44]. In many applications, it is not clear which performance measure the demonstrator is (possibly subconsciously) optimizing. That is, the reward signal is not labeled and accessible in the observed expert's trajectories. In such settings, the performance of candidate policies is not uniquely discernible from the observational data due to latent outcomes, even when infinitely many samples are gathered, complicating efforts to learn policy with satisfactory performance.

An alternative approach used to circumvent this issue is to find a policy that mimics a demonstrator's behavior, which leads to the *imitation learning* paradigm [2, 4, 14, 28]. The expectation (or rather hope) is that if the demonstrations are generated by an expert with near-optimal reward, the performance of the imitator would also be satisfactory. Current methods of imitation learning can be categorized into *behavior cloning* [45, 31, 23, 24, 22] and *inverse reinforcement learning* [25, 1, 38, 46]. The former focuses on learning a nominal expert policy that approximates the conditional distribution mapping observed input covariates of the behavior policy to the action domain. The latter attempts to learn a reward function that prioritizes observed behaviors of the expert; reinforcement learning methods are then applied using the learned reward function to obtain a nominal

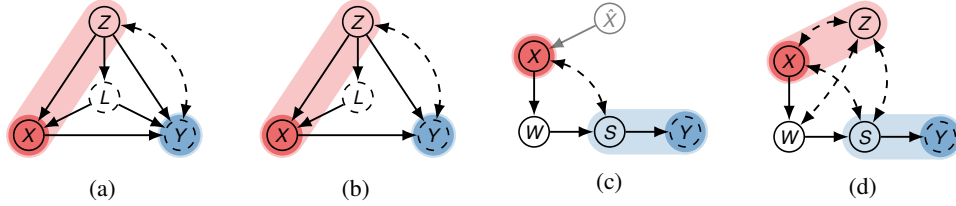

(a)  (b)  (c)  (d)

Figure 1: Causal diagrams where $X$ represents an action (shaded red) and $Y$ represents a latent reward (shaded blue). Input covariates of the policy space $\Pi$ are shaded in light red and minimal imitation surrogates relative to action $X$ and reward $Y$ are shaded in light blue.

policy. However, both families of methods rely on the assumption that the expert's input observations match those available to the imitator. When unobserved covariates exist, however, naively imitating the nominal expert policy does not necessarily lead to a satisfactory performance, even when the expert him or herself behaves optimally.

For concreteness, consider a learning scenario depicted in Fig. 2, describing trajectories of human-driven cars collected by drones flying over highways [18, 8]. Using such data, we want to learn a policy $\pi(x|z)$ deciding on the acceleration (action) $X$ of the demonstrator car based on the velocity and locations of both the demonstrator and front cars, summarized as covariates $Z$. In reality, the human demonstrator also uses the tail light $L$ of the front car to coordinate his/her actions. The demonstrator's performance is evaluated with a latent reward function $Y$ taking $X, Z, L$ as input. However, only observations of $X, Z$ are collected by the drone, summarized as

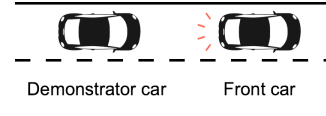

Demonstrator car  Front car

Figure 2: The tail light of the front car is unobserved in highway (aerial) drone data.

probabilities $P(x, z)$. Fig. 1a describes the graphical representation of this environment. A naïve approach would estimate the conditional distribution $P(x|z)$ and use it as policy $\pi$. A preliminary analysis reveals that this naive "cloning" approach leads to sub-optimal performance. Consider an instance where variables $X, Y, Z, L, U \in \{0, 1\}$; their values are decided by functions: $L \leftarrow Z \oplus U$, $X \leftarrow Z \oplus \neg L$, $Y \leftarrow X \oplus Z \oplus L$; $Z, U$ are independent variables drawn uniformly over $\{0, 1\}$; $\oplus$ represents the *exclusive-or* operator. The expected reward $\mathbb{E}[Y|\text{do}(\pi)]$ induced by $\pi(x|z) = P(x|z)$ is equal to $0.5$, which is quite far from the optimal demonstrator's performance, $\mathbb{E}[Y] = 1$.

This example shows that even when one is able to perfectly mimic an optimal demonstrator, the learned policy can still be suboptimal. In this paper, we try to explicate this phenomenon and, more broadly, understand *imitability* through a causal lens[1]. Our task is to learn an imitating policy that achieves the expert's performance from demonstration data in a *structural causal model* [29, Ch. 7], allowing for unobserved confounders (UCs) affecting both action and outcome variables. Specifically, our contributions are summarized as follows. (1) We introduce a complete graphical criterion for determining the feasibility of imitation from demonstration data and qualitative knowledge about the data-generating process represented as a causal graph. (2) We develop a sufficient algorithm for identifying an imitating policy when the given criterion does not hold, by leveraging the quantitative knowledge in the observational distribution. (3) We provide an efficient and practical procedure for finding an imitating policy through explicit parametrization of the causal model, and use it to validate our results on high-dimensional, synthetic datasets. For the sake of space constraints, we provide all proofs in the complete technical report [15, Appendix A].

## 1.1 Preliminaries

In this section, we introduce the basic notations and definitions used throughout the paper. We use capital letters to denote random variables ($X$) and small letters for their values ($x$). $\mathscr{D}_X$ represents the domain of $X$ and $\mathscr{P}_X$ the space of probability distributions over $\mathscr{D}_X$. For a set $\boldsymbol{X}$, $|\boldsymbol{X}|$ denotes its dimension. We consistently use the abbreviation $P(x)$ to represent the probabilities $P(X = x)$. Finally, $I_{\{\boldsymbol{Z}=\boldsymbol{z}\}}$ is an indicator function that returns $1$ if $\boldsymbol{Z} = \boldsymbol{z}$ holds true; otherwise $0$.

Calligraphic letters, e.g., $\mathcal{G}$, will be used to represent directed acyclic graphs (DAGs) (e.g., Fig. 1). We denote by $\mathcal{G}_{\overline{X}}$ the subgraph obtained from $\mathcal{G}$ by removing arrows coming into nodes in $X$; $\mathcal{G}_{\underline{X}}$ is a subgraph of $\mathcal{G}$ by removing arrows going out of $X$. We will use standard family conventions for graphical relationships such as parents, children, descendants, and ancestors. For example, the set of parents of $X$ in $\mathcal{G}$ is denoted by $pa(X)_{\mathcal{G}} = \cup_{X \in X} pa(X)_{\mathcal{G}}$. $ch$, $de$ and $an$ are similarly defined, We write $Pa, Ch, De, An$ if arguments are included as well, e.g. $De(X)_{\mathcal{G}} = de(X)_{\mathcal{G}} \cup X$. A path from a node $X$ to a node $Y$ in $\mathcal{G}$ is a sequence of edges which does not include a particular node more than once. Two sets of nodes $X, Y$ are said to be d-separated by a third set $Z$ in a DAG $\mathcal{G}$, denoted by $(X \perp\!\!\!\perp Y | Z)_{\mathcal{G}}$, if every edge path from nodes in one set to nodes in another are "blocked". The criterion of blockage follows [29, Def. 1.2.3].

The basic semantic framework of our analysis rests on *structural causal models* (SCMs) [29, Ch. 7]. An SCM $M$ is a tuple $\langle U, V, \mathcal{F}, P(u) \rangle$ where $V$ is a set of endogenous variables and $U$ is a set of exogenous variables. $\mathcal{F}$ is a set of structural functions where $f_V \in \mathcal{F}$ decides values of an endogenous variable $V \in V$ taking as argument a combination of other variables. That is, $V \leftarrow f_V(Pa_V, U_V), Pa_V \subseteq V, U_V \subseteq U$. Values of $U$ are drawn from an exogenous distribution $P(u)$. Each SCM $M$ induces a distribution $P(v)$ over endogenous variables $V$. An intervention on a subset $X \subseteq V$, denoted by $do(x)$, is an operation where values of $X$ are set to constants $x$, replacing the functions $\{f_X : \forall X \in X\}$ that would normally determine their values. For an SCM $M$, let $M_x$ be a submodel of $M$ induced by intervention $do(x)$. For a set $S \subseteq V$, the interventional distribution $P(s|do(x))$ induced by $do(x)$ is defined as the distribution over $S$ in the submodel $M_x$, i.e., $P(s|do(x); M) \triangleq P(s; M_x)$. We leave $M$ implicit when it is obvious from the context. For a detailed survey on SCMs, we refer readers to [29, Ch. 7].

## 2 Imitation Learning in Structural Causal Models

In this section, we formalize and study the imitation learning problem in causal language. We first define a special type of SCM that explicitly allows one to model the unobserved nature of some endogenous variables, which is called the partially observable structural causal model (POSCM).[2]

**Definition 1** (Partially Observable SCM). A POSCM is a tuple $\langle M, O, L \rangle$, where $M$ is a SCM $\langle U, V, \mathcal{F}, P(u) \rangle$ and $\langle O, L \rangle$ is a pair of subsets forming a partition over $V$ (i.e., $V = O \cup L$ and $O \cap L = \emptyset$); $O$ and $L$ are called observed and latent endogenous variables, respectively.

Each POSCM $M$ induces a probability distribution over $V$, of which one can measure the observed variables $O$. $P(o)$ is usually called the *observational* distribution. $M$ is associated with a *causal diagram* $\mathcal{G}$ (e.g., see Fig. 1) where solid nodes represent observed variables $O$, dashed nodes represent latent variables $L$, and arrows represent the arguments $Pa_V$ of each functional relationship $f_V$. Exogenous variables $U$ are not explicitly shown; a bi-directed arrow between nodes $V_i$ and $V_j$ indicates the presence of an unobserved confounder (UC) affecting both $V_i$ and $V_j$, i.e., $U_{V_i} \cap U_{V_j} \neq \emptyset$.

Consider a POSCM $\langle M, O, L \rangle$ with $M = \langle U, V, \mathcal{F}, P(u) \rangle$. Our goal is to learn an efficient policy to decide the value of an action variable $X \in O$. The performance of the policy is evaluated using the expected value of a reward variable $Y$. Throughout this paper, we assume that reward $Y$ is latent and $X$ affects $Y$ (i.e., $Y \in L \cap De(X)_{\mathcal{G}}$). A *policy* $\pi$ is a function mapping from values of covariates $Pa^* \subseteq O \setminus De(X)_{\mathcal{G}_{\overline{X}}}$[3] to a probability distribution over $X$, which we denote by $\pi(x|pa^*)$. An intervention following a policy $\pi$, denoted by $do(\pi)$, is an operation that draws values of X independently following $\pi$, regardless of its original (natural) function $f_X$. Let $M_\pi$ denote the manipulated SCM of $M$ induced by $do(\pi)$. Similar to atomic settings, the interventional distribution $P(v|do(\pi))$ is defined as the distribution over $V$ in the manipulated model $M_\pi$, given by,

$$P(v|do(\pi)) = \sum_u P(u) \prod_{V \in V \setminus \{X\}} P(v|pa_V, u_V) \pi(x|pa^*). \tag{1}$$

The expected reward of a policy $\pi$ is thus given by the causal effect $\mathbb{E}[Y|do(\pi)]$. The collection of all possible policies $\pi$ defines a *policy space*, denoted by $\Pi = \{\pi : \mathscr{D}_{Pa^*} \mapsto \mathscr{P}_X\}$ (if $Pa^* = \emptyset$, $\Pi = \{\pi : \mathscr{P}_X\}$). For convenience, we define function $Pa(\Pi) = Pa^*$. A policy space $\Pi'$ is a

subspace of $\Pi$ if $Pa(\Pi') \subseteq Pa(\Pi)$. We will consistently highlight action $X$ in dark red, reward $Y$ in dark blue and covariates $Pa(\Pi)$ in light red. For instance, in Fig. 1a, the policy space over action $X$ is given by $\Pi = \{\pi : \mathscr{D}_Z \mapsto \mathscr{P}_X\}$; $Y$ represents the reward; $\Pi' = \{\pi : \mathscr{P}_X\}$ is a subspace of $\Pi$.

Our goal is to learn an efficient policy $\pi \in \Pi$ that achieves satisfactory performance, e.g., larger than a certain threshold $\mathbb{E}[Y|\mathrm{do}(\pi)] \geq \tau$, without knowledge of underlying system dynamics, i.e., the actual, true POSCM $M$. A possible approach is to identify the expected reward $\mathbb{E}[Y|\mathrm{do}(\pi)]$ for each policy $\pi \in \Pi$ from the combinations of the observed data $P(\boldsymbol{o})$ and the causal diagram $\mathcal{G}$. Optimization procedures are applicable to find a satisfactory policy $\pi$. Let $\mathscr{M}_{\langle\mathcal{G}\rangle}$ denote a hypothesis class of POSCMs that are compatible with a causal diagram $\mathcal{G}$. We define the non-parametric notion of identifiability in the context of POSCMs and conditional policies, adapted from [29, Def. 3.2.4].

**Definition 2** (Identifiability). Given a causal diagram $\mathcal{G}$ and a policy space $\Pi$, let $\boldsymbol{Y}$ be an arbitrary subset of $\boldsymbol{V}$. $P(\boldsymbol{y}|\mathrm{do}(\pi))$ is said to be identifiable w.r.t. $\langle\mathcal{G}, \Pi\rangle$ if $P(\boldsymbol{y}|\mathrm{do}(\pi); M)$ is uniquely computable from $P(\boldsymbol{o}; M)$ and $\pi$ for any POSCM $M \in \mathscr{M}_{\langle\mathcal{G}\rangle}$ and any $\pi \in \Pi$.

In imitation learning settings, however, reward $Y$ is often not specified and remains latent, which precludes approaches that attempt to identify $\mathbb{E}[Y|\mathrm{do}(\pi)]$:

**Corollary 1.** *Given a causal diagram $\mathcal{G}$ and a policy space $\Pi$, let $\boldsymbol{Y}$ be an arbitrary subset of $\boldsymbol{V}$. If not all variables in $\boldsymbol{Y}$ are observed (i.e., $\boldsymbol{Y} \cap \boldsymbol{L} \neq \emptyset$), $P(\boldsymbol{y}|\mathrm{do}(\pi))$ is not identifiable.*

In other words, Corol. 1 shows that when the reward $Y$ is latent, it is infeasible to uniquely determine values of $\mathbb{E}[Y|\mathrm{do}(\pi)]$ from $P(\boldsymbol{o})$. A similar observation has been noted in [20, Prop. 1]. This suggests that we need to explore learning through other modalities.

## 2.1 Causal Imitation Learning

To circumvent issues of non-identifiability, a common solution is to assume that the observed trajectories are generated by an "expert" demonstrator with satisfactory performance $\mathbb{E}[Y]$, e.g., no less than a certain threshold ($\mathbb{E}[Y] \geq \tau$). If we could find a policy $\pi$ that perfectly "imitates" the expert with respect to reward $Y$, $\mathbb{E}[Y|\mathrm{do}(\pi)] = \mathbb{E}[Y]$, the performance of the learner is also guaranteed to be satisfactory. Formally,

**Definition 3** (Imitability). Given a causal diagram $\mathcal{G}$ and a policy space $\Pi$, let $\boldsymbol{Y}$ be an arbitrary subset of $\boldsymbol{V}$. $P(\boldsymbol{y})$ is said to be imitable w.r.t. $\langle\mathcal{G}, \Pi\rangle$ if there exists a policy $\pi \in \Pi$ uniquely computable from $P(\boldsymbol{o})$ such that $P(\boldsymbol{y}|\mathrm{do}(\pi); M) = P(\boldsymbol{y}; M)$ for any POSCM $M \in \mathscr{M}_{\langle\mathcal{G}\rangle}$.

Our task is to determine the imitability of the expert performance. More specifically, we want to learn an *imitating policy* $\pi \in \Pi$ from $P(\boldsymbol{o})$ such that $P(y|\mathrm{do}(\pi)) = P(y)$[4], in any POSCM $M$ associated with the causal diagram $\mathcal{G}$. Consider Fig. 3a as an example. $P(y)$ is imitable with policy $\pi(x) = P(x)$ since by Eq. (1) and marginalization, $P(y|\mathrm{do}(\pi)) = \sum_{x,w} P(y|w)P(w|x)\pi(x) = \sum_{x,w} P(y|w)P(w|x)P(x) = P(y)$. In practice, unfortunately, the expert's performance cannot always be imitated. To understand this setting, we first write, more explicitly, the conditions under which this is not the case:

**Lemma 1.** *Given a causal diagram $\mathcal{G}$ and a policy space $\Pi$, let $\boldsymbol{Y}$ be an arbitrary subset of $\boldsymbol{V}$. $P(\boldsymbol{y})$ is not imitable w.r.t. $\langle\mathcal{G}, \Pi\rangle$ if there exists two POSCMs $M_1, M_2 \in \mathscr{M}_{\langle\mathcal{G}\rangle}$ satisfying $P(\boldsymbol{o}; M_1) = P(\boldsymbol{o}; M_2)$ while there exists no policy $\pi \in \Pi$ such that for $i = 1, 2$, $P(\boldsymbol{y}|\mathrm{do}(\pi); M_i) = P(\boldsymbol{y}; M_i)$.*

It follows as a corollary that $P(\boldsymbol{y})$ is not imitable if there exists a POSCM $M$ compatible with $\mathcal{G}$ such that no policy $\pi \in \Pi$ could ensure $P(\boldsymbol{y}|\mathrm{do}(\pi); M) = P(\boldsymbol{y}; M)$. For instance, consider the causal diagram $\mathcal{G}$ and policy space $\Pi$ in Fig. 3b. Here, the expert's reward $P(y)$ is not imitable: consider a POSCM with functions $X \leftarrow U, W \leftarrow X, Y \leftarrow U \oplus \neg W$; values $U$ are drawn uniformly over $\{0, 1\}$. In this model, $P(Y = 1|\mathrm{do}(\pi)) = 0.5$ for any policy $\pi$, which is far from the optimal expert reward, $P(Y = 1) = 1$.

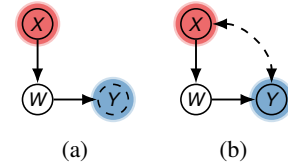

(a)        (b)

Figure 3: Imitability v. Identifiability.

An interesting observation from the above example of Fig. 3b is that the effect $P(y|\mathrm{do}(\pi))$ is identifiable, following the front-door criterion in [29, Thm. 3.3.4], but no

policy imitates the corresponding $P(y)$. However, in some settings, the expert's reward $P(y)$ is imitable but the imitator's reward $P(y|\text{do}(\pi))$ cannot be uniquely determined. To witness, consider again the example in Fig. 3a. The imitability of $P(y)$ has been previously shown; while $P(y|\text{do}(\pi))$ is not identifiable due to latent reward $Y$ (Corol. 1).

In general, the problem of imitability is orthogonal to identifiability, and, therefore, requires separate consideration. Since imitability does not always hold, we introduce a useful graphical criterion for determining whether imitating an expert's performance is feasible, and if so, how.

**Theorem 1** (Imitation by Direct Parents). *Given a causal diagram $\mathcal{G}$ and a policy space $\Pi$, $P(y)$ is imitable w.t.r. $\langle \mathcal{G}, \Pi \rangle$ if $pa(X)_{\mathcal{G}} \subseteq Pa(\Pi)$ and there is no bi-directed arrow pointing to $X$ in $\mathcal{G}$. Moreoever, the imitating policy $\pi \in \Pi$ is given by $\pi(x|pa(\Pi)) = P(x|pa(X)_{\mathcal{G}})$.*

In words, Thm. 1 says that if the expert and learner share the same policy space, then the policy is always imitable. In fact, this result can be seen as a causal justification for when the method of "behavior cloning", widely used in practice, is valid, leading to proper imitation. When the original behavior policy $f_X$ is contained in the policy space $\Pi$, the leaner could imitate the expert's reward $P(y)$ by learning a policy $\pi \in \Pi$ that matches the distribution $P(x|pa(\Pi))$ [45, 31]. Next, we consider the more challenging setting when policy spaces of the expert and learner disagree (i.e., the learner and expert have different views of the world, $f_X \notin \Pi$). We will leverage a graphical condition adopted from the celebrated *backdoor criterion* [29, Def. 3.3.1].

**Definition 4** ($\pi$-Backdoor). Given a causal diagram $\mathcal{G}$ and a policy space $\Pi$, a set $\mathbf{Z}$ is said to satisfy the $\pi$-*backdoor* criterion w.r.t. $\langle \mathcal{G}.\Pi \rangle$ if and only if $\mathbf{Z} \subseteq Pa(\Pi)$ and $(Y \perp\!\!\!\perp X|\mathbf{Z})_{\mathcal{G}_{\underline{X}}}$, which is called the $\pi$-*backdoor admissible set* w.r.t. $\langle \mathcal{G}, \Pi \rangle$.

For concreteness, consider again the highway driving example in Fig. 1a. There exists no $\pi$-backdoor admissible set due to the path $X \leftarrow L \rightarrow Y$. Now consider a modified graph in Fig. 1b where edge $L \rightarrow Y$ is removed. $\{Z\}$ is $\pi$-backdoor admissible since $Z \in Pa(\Pi)$ and $(Y \perp\!\!\!\perp X|Z)_{\mathcal{G}_{\underline{X}}}$. Leveraging the imitation backdoor condition, our next theorem provides a full characterization for when imitating expert's performance is achievable, despite the fact that the reward $Y$ is latent.

**Theorem 2** (Imitation by $\pi$-Backdoor). *Given a causal diagram $\mathcal{G}$ and a policy space $\Pi$, $P(y)$ is imitable w.r.t. $\langle \mathcal{G}, \Pi \rangle$ if and only if there exists an $\pi$-backdoor admissible set $\mathbf{Z}$ w.r.t. $\langle \mathcal{G}, \Pi \rangle$. Moreover, the imitating policy $\pi \in \Pi$ is given by $\pi(x|pa(\Pi)) = P(x|\mathbf{z})$.*

That is, one can learn an imitating policy from a policy space $\Pi' = \{\pi : \mathscr{D}_{\mathbf{Z}} \mapsto \mathscr{P}_X\}$ that mimics the conditional probabilities $P(x|\mathbf{z})$ if and only if $\mathbf{Z}$ is $\pi$-backdoor admissible. If that is the case, such a policy can be learned from data through standard density estimation methods. For instance, Thm. 2 ascertains that $P(y)$ in Fig. 1a is indeed non-imitable. On the other hand, $P(y)$ in Fig. 1b is imitable, guaranteed by the $\pi$-backdoor admissible set $\{Z\}$; the imitating policy is given by $\pi(x|z) = P(x|z)$.

## 3   Causal Imitation Learning with Data Dependency

One may surmise that the imitation boundary established by Thm. 2 suggests that when there exists no $\pi$-backdoor admissible set, it is infeasible to imitate the expert performance from observed trajectories of demonstrations. In this section, we will circumvent this issue by exploiting actual parameters of the observational distribution $P(\mathbf{o})$. In particular, we denote by $\mathscr{M}_{\langle \mathcal{G}, P \rangle}$ a subfamily of candidate models in $\mathscr{M}_{\langle \mathcal{G} \rangle}$ that induce both the causal diagram $\mathcal{G}$ and the observational distribution $P(\mathbf{o})$, i.e., $\mathscr{M}_{\langle \mathcal{G}, P \rangle} = \{\forall M \in \mathscr{M}_{\langle \mathcal{G} \rangle} : P(\mathbf{o}; M) = P(\mathbf{o})\}$. We introduce a refined notion of imitability that will explore the quantitative knowledge of observations $P(\mathbf{o})$ (to be exemplified). Formally,

**Definition 5** (Practical Imitability). Given a causal diagram $\mathcal{G}$, a policy space $\Pi$, and an observational distribution $P(\mathbf{o})$, let $\mathbf{Y}$ be an arbitrary subset of $\mathbf{V}$. $P(\mathbf{y})$ is said to be *practically imitable* (for short, p-imitable) w.r.t. $\langle \mathcal{G}, \Pi, P(\mathbf{o}) \rangle$ if there exists a policy $\pi \in \Pi$ uniquely computable from $P(\mathbf{o})$ such that $P(\mathbf{y}|\text{do}(\pi); M) = P(\mathbf{y}; M)$ for any POSCM $M \in \mathscr{M}_{\langle \mathcal{G}, P \rangle}$.

The following corollary can be derived based on the definition of practical imitability.

**Corollary 2.** *Given a causal diagram $\mathcal{G}$, a policy space $\Pi$ and an observational distribution $P(\mathbf{o})$, let a subset $\mathbf{Y} \subseteq \mathbf{V}$. If $P(\mathbf{y})$ is imitable w.r.t. $\langle \mathcal{G}, \Pi \rangle$, $P(\mathbf{y})$ is p-imitable w.r.t. $\langle \mathcal{G}, \Pi, P(\mathbf{o}) \rangle$.*

Compared to Def. 3, the practical imitability of Def. 5 aims to find an imitating policy for a subset of candidate POSCMs $\mathscr{M}_{\langle \mathcal{G}, P \rangle}$ restricted to match a specific observational distribution $P(\mathbf{o})$. Def. 3, on

the other hand, requires only the causal diagram $\mathcal{G}$. In other words, for an expert's performance $P(y)$ that is non-imitable w.r.t. $\langle \mathcal{G}, \Pi \rangle$, it could still be p-imitable after analyzing actual probabilities of the observational distribution $P(\boldsymbol{o})$.

For concreteness, consider again $P(y)$ in Fig. 3b which is not imitable due to the bi-directed arrow $X \leftrightarrow Y$. However, new imitation opportunities arise when actual parameters of the observational distribution $P(x, w, y)$ are provided. Suppose the underlying POSCM is given by: $X \leftarrow U_X \oplus U_Y$, $W \leftarrow X \oplus U_W$, $Y \leftarrow W \oplus U_Y$ where $U_X, U_Y, U_W$ are independent binary variables drawn from $P(U_X = 1) = P(U_Y = 1) = P(U_W = 0) = 0.9$. Here, the causal effect $P(y|\mathrm{do}(x))$ is identifiable from $P(x, w, y)$ following the front-door formula $P(y|\mathrm{do}(x)) = \sum_w P(w|x) \sum_{x'} P(y|w, x')P(x')$ [29, Thm. 3.3.4]. We thus have $P(Y = 1|\mathrm{do}(X = 0)) = 0.82$ which coincides with $P(Y = 1) = 0.82$, i.e., $P(y)$ is p-imitable with atomic intervention $\mathrm{do}(X = 0)$. In the most practical settings, the expert reward $P(y)$ rarely equates to $P(y|\mathrm{do}(x))$; stochastic policies $\pi(x)$ are then applicable to imitate $P(y)$ by re-weighting $P(y|\mathrm{do}(x))$ induced by the corresponding atomic interventions [5]. To tackle p-imitability in a general way, we proceed by defining a set of observed variables that serve as a surrogate of the unobserved $Y$ with respect to interventions on $X$. Formally,

**Definition 6** (Imitation Surrogate). Given a causal diagram $\mathcal{G}$, a policy space $\Pi$, let $\boldsymbol{S}$ be an arbitrary subset of $\boldsymbol{O}$. $\boldsymbol{S}$ is an *imitation surrogate* (for short, surrogate) w.r.t. $\langle \mathcal{G}, \Pi \rangle$ if $(Y \perp\!\!\!\perp \hat{X}|\boldsymbol{S})_{\mathcal{G} \cup \Pi}$ where $\mathcal{G} \cup \Pi$ is a supergraph of $\mathcal{G}$ by adding arrows from $Pa(\Pi)$ to $X$; $\hat{X}$ is a new parent to $X$.

An surrogate $\boldsymbol{S}$ is said to be minimal if there exists no subset $\boldsymbol{S}' \subset \boldsymbol{S}$ such that $\boldsymbol{S}'$ is also a surrogate w.r.t. $\langle \mathcal{G}, \Pi \rangle$. Consider as an example Fig. 1c where the supergraph $\mathcal{G} \cup \Pi$ coincides with the causal diagram $\mathcal{G}$. By Def. 6, both $\{W, S\}$ and $\{S\}$ are valid surrogate relative to $\langle X, Y \rangle$ with $\{S\}$ being the minimal one. By conditioning on $S$, the decomposition of Eq. (1) implies $P(y|\mathrm{do}(\pi)) = \sum_{s,w,u} P(y|s)P(s|w, u)P(w|x)\pi(x)P(u) = \sum_s P(y|s)P(s|\mathrm{do}(\pi))$. That is, the surrogate $S$ mediates all influence of interventions on action $X$ to reward $Y$. It is thus sufficient to find an imitating policy $\pi$ such that $P(s|\mathrm{do}(\pi)) = P(s)$ for any POSCM $M$ associated with Fig. 1c. The resultant policy is guaranteed to imitate the expert's reward $P(y)$.

When a surrogate $\boldsymbol{S}$ is found and $P(\boldsymbol{s}|\mathrm{do}(\pi))$ is identifiable, one could compute $P(\boldsymbol{s}|\mathrm{do}(\pi))$ for each policy $\pi$ and check if it matches $P(\boldsymbol{s})$. In many settings, however, $P(\boldsymbol{s}|\mathrm{do}(\pi))$ is not identifiable w.r.t. $\langle \mathcal{G}, \Pi \rangle$. For example, in Fig. 1d, $S$ is a surrogate w.r.t. $\langle \mathcal{G}, \Pi \rangle$, but $P(s|\mathrm{do}(\pi))$ is not identifiable due to collider $Z$ ($\pi$ uses non-descendants as input by default). Fortunately, identifying $P(\boldsymbol{s}|\mathrm{do}(\pi))$ may still be feasible in some subspaces of $\Pi$:

**Definition 7** (Identifiable Subspace). Given a causal diagram $\mathcal{G}$, a policy space $\Pi$, and a subset $\boldsymbol{S} \subseteq \boldsymbol{O}$, let $\Pi'$ be a policy subspace of $\Pi$. $\Pi'$ is said to be an *identifiable subspace* (for short, id-subspace) w.r.t. $\langle \mathcal{G}, \Pi, \boldsymbol{S} \rangle$ if $P(\boldsymbol{s}|\mathrm{do}(\pi))$ is identifiable w.r.t. $\langle \mathcal{G}, \Pi' \rangle$.

Consider a policy subspace $\Pi' = \{\pi : \mathscr{P}_X\}$ described in Fig. 1d (i.e. $\pi$ that does not exploit information from covariates $Z$). $P(s|\mathrm{do}(\pi))$ is identifiable w.r.t. $\langle \mathcal{G}, \Pi' \rangle$ following the front-door adjustment on $W$ [29, Thm. 3.3.4]. We could then evaluate interventional probabilities $P(s|\mathrm{do}(\pi))$ for each policy $\pi \in \Pi'$ from the observational distribution $P(x, w, s, z)$; the imitating policy is obtainable by solving the equation $P(s|\mathrm{do}(\pi)) = P(s)$. In other words, $\{S\}$ and $\Pi'$ forms an instrument that allows one to solve the imitation learning problem in Fig. 1d.

**Definition 8** (Imitation Instrument). Given a causal diagram $\mathcal{G}$ and a policy space $\Pi$, let $\boldsymbol{S}$ be a subset of $\boldsymbol{O}$ and $\Pi'$ be a subspace of $\Pi$. $\langle \boldsymbol{S}, \Pi' \rangle$ is said to be an *imitation instrument* (for short, instrument) if $\boldsymbol{S}$ is a surrogate w.r.t. $\langle \mathcal{G}, \Pi' \rangle$ and $\Pi'$ is an id-subspace w.r.t. $\langle \mathcal{G}, \Pi, \boldsymbol{S} \rangle$.

**Lemma 2.** *Given a causal diagram $\mathcal{G}$, a policy space $\Pi$, and an observational distribution $P(\boldsymbol{o})$, let $\langle \boldsymbol{S}, \Pi' \rangle$ be an instrument w.r.t. $\langle \mathcal{G}, \Pi \rangle$. If $P(\boldsymbol{s})$ is p-imitable w.r.t. $\langle \mathcal{G}, \Pi', P(\boldsymbol{o}) \rangle$, then $P(y)$ is p-imitable w.r.t. $\langle \mathcal{G}, \Pi, P(\boldsymbol{o}) \rangle$. Moreover, an imitating policy $\pi$ for $P(\boldsymbol{s})$ w.r.t. $\langle \mathcal{G}, \Pi', P(\boldsymbol{o}) \rangle$ is also imitating policy for $P(y)$ w.r.t. $\langle \mathcal{G}, \Pi, P(\boldsymbol{o}) \rangle$.*

In words, Lem. 2 shows that when an imitation instrument $\langle \boldsymbol{S}, \Pi' \rangle$ is present, we could reduce the original imitation learning on a latent reward $Y$ to a p-imitability problem over observed surrogate variables $\boldsymbol{S}$ using policies in an identifiable subspace $\Pi'$. The imitating policy $\pi$ is obtainable by solving the equation $P(\boldsymbol{s}|\mathrm{do}(\pi)) = P(\boldsymbol{s})$.

## 3.1 Confounding Robust Imitation

Our task in this section is to introduce a general algorithm that finds instruments, and learns a p-imitating policy given $\langle \mathcal{G}, \Pi, P(\boldsymbol{o}) \rangle$. A naïve approach is to enumerate all pairs of subset $\boldsymbol{S}$ and subspace $\Pi'$ and check whether they form an instrument; if so, we can compute an imitating policy for $P(\boldsymbol{s})$ w.r.t. $\langle \mathcal{G}, \Pi', P(\boldsymbol{o}) \rangle$. However, the challenge is that the number of all possible subspaces $\Pi'$ (or subsets $\boldsymbol{S}$) can be exponentially large. Fortunately, we can greatly restrict this search space. Let $\mathcal{G} \cup \{Y\}$ denote a causal diagram obtained from $\mathcal{G}$ by making reward $Y$ observed. The following proposition suggests that it suffices to consider only identifiable subspaces w.r.t. $\langle \mathcal{G} \cup \{Y\}, \Pi, Y \rangle$.

**Lemma 3.** *Given a causal diagram $\mathcal{G}$, a policy space $\Pi$, let a subspace $\Pi' \subseteq \Pi$. If there exists $\boldsymbol{S} \subseteq \boldsymbol{O}$ such that $\langle \boldsymbol{S}, \Pi' \rangle$ is an instrument w.r.t. $\langle \mathcal{G}, \Pi \rangle$, $\Pi'$ is an id-subspace w.r.t. $\langle \mathcal{G} \cup \{Y\}, \Pi, Y \rangle$.*

Our algorithm IMITATE is described in Alg. 1. We assume access to an IDEN-TIFY oracle [41, 34, 6] that takes as input a causal diagram $\mathcal{G}$, a policy space $\Pi$ and a set of observed variables $\boldsymbol{S}$. If $P(\boldsymbol{s}|\text{do}(\pi))$ is identifiable w.r.t. $\langle \mathcal{G}, \Pi \rangle$, IDENTIFY returns "YES"; otherwise, it returns "NO". For details about the IDEN-TIFY oracle, we refer readers to [15, Appendix B]. More specifically, IMITATE takes as input a causal diagram $\mathcal{G}$, a policy space $\Pi$ and an observational distribution $P(\boldsymbol{o})$. At Step 2, IMITATE applies a subroutine LISTIDSPACE to list identifiable subspaces $\Pi'$ w.r.t. $\langle \mathcal{G} \cup \{Y\}, \Pi, Y \rangle$, following the observation made in Lem. 3. The implementation details of LISTIDSPACE are

---

**Algorithm 1:** IMITATE

1: **Input:** $\mathcal{G}, \Pi, P(\boldsymbol{o})$.
2: **while** LISTIDSPACE$(\mathcal{G} \cup \{Y\}, \Pi, Y)$ outputs a policy subspace $\Pi'$ **do**
3:      **while** LISTMINSEP$(\mathcal{G} \cup \Pi', \hat{X}, Y, \{\}, \boldsymbol{O})$ outputs a surrogate set $\boldsymbol{S}$ **do**
4:          **if** IDENTIFY$(\mathcal{G}, \Pi', \boldsymbol{S}) =$ YES **then**
5:              Solve for a policy $\pi \in \Pi'$ such that
$$P(\boldsymbol{s}|\text{do}(\pi); M) = P(\boldsymbol{s})$$
             for any POSCM $M \in \mathcal{M}_{\langle \mathcal{G}, P \rangle}$.
6:              Return $\pi$ if it exists; continue otherwise.
7:          **end if**
8:      **end while**
9: **end while** Return FAIL.

---

provided in [15, Appendix C]. When an identifiable subspace $\Pi'$ is found, IMITATE tries to obtain a surrogate $\boldsymbol{S}$ w.r.t the diagram $\mathcal{G}$ and subspace $\Pi'$. While there could exist multiple such surrogates, the following proposition shows that it is sufficient to consider only minimal ones.

**Lemma 4.** *Given a causal diagram $\mathcal{G}$, a policy space $\Pi$, an observational distribution $P(\boldsymbol{o})$ and a subset $\boldsymbol{S} \subseteq \boldsymbol{O}$. $P(\boldsymbol{s})$ is p-imitable only if for any $\boldsymbol{S}' \subseteq \boldsymbol{S}$, $P(\boldsymbol{s}')$ is p-imitable w.r.t. $\langle \mathcal{G}, \Pi, P(\boldsymbol{o}) \rangle$.*

We apply a subroutine LISTMINSEP in [43] to enumerate minimal surrogates in $\boldsymbol{O}$ that d-separate $\hat{X}$ and $Y$ in the supergraph $\mathcal{G} \cup \Pi'$. When a minimal surrogate $\boldsymbol{S}$ is found, IMITATE uses the IDENTIFY oracle to validate if $P(\boldsymbol{s}|\text{do}(\pi))$ is identifiable w.r.t. $\langle \mathcal{G}, \Pi' \rangle$, i.e., $\langle \boldsymbol{S}, \Pi' \rangle$ form an instrument. Consider Fig. 1d as an example. While $P(y|\text{do}(\pi))$ is not identifiable for every policy in $\Pi$ had $Y$ been observed, $\Pi$ contains an id-subspace $\{\pi : \mathscr{P}_X\}$ w.r.t. $\langle \mathcal{G} \cup \{Y\}, \Pi, Y \rangle$, which is associated with a minimal surrogate $\{S\}$. Applying IDENTIFY confirms that $\langle \{S\}, \{\pi : \mathscr{P}_X\} \rangle$ is an instrument.

At Step 5, IMITATE solves for a policy $\pi$ in the subspace $\Pi'$ that imitates $P(\boldsymbol{s})$ for all instances in the hypothesis class $\mathcal{M}_{\langle \mathcal{G}, P \rangle}$. If such a policy exists, IMITATE returns $\pi$; otherwise, the algorithm continues. Since $\langle \boldsymbol{S}, \Pi' \rangle$ is an instrument, Lem. 2 implies that the learned policy $\pi$, if it exists, is ensured to imitate the expert reward $P(y)$ for any POSCM $M \in \mathcal{M}_{\langle \mathcal{G}, P \rangle}$.

**Theorem 3.** *Given a causal diagram $\mathcal{G}$, a policy space $\Pi$, and an observational distribution $P(\boldsymbol{o})$, if IMITATE returns a policy $\pi \in \Pi$, $P(y)$ is p-imitable w.r.t. $\langle \mathcal{G}, \Pi, P(\boldsymbol{o}) \rangle$. Moreover, $\pi$ is an imitating policy for $P(y)$ w.r.t. $\langle \mathcal{G}, \Pi, P(\boldsymbol{o}) \rangle$.*

## 3.2 Optimizing Imitating Policies

We now introduce optimization procedures to solve for an imitating policy at Step 5 of IMITATE algorithm. Since the pair $\langle \boldsymbol{S}, \Pi' \rangle$ forms a valid instrument (ensured by Step 4), the interventional distribution $P(\boldsymbol{s}|\text{do}(\pi); M)$ remains invariant among all models in $\mathcal{M}_{\langle \mathcal{G} \rangle}$, i.e., $P(\boldsymbol{s}|\text{do}(\pi))$ is identifiable w.r.t. $\langle \mathcal{G}, \Pi \rangle$. We could thus express $P(\boldsymbol{s}|\text{do}(\pi); M)$ for any $M \in \mathcal{M}_{\langle \mathcal{G}, P \rangle}$ as a function of the observational distribution $P(\boldsymbol{o})$; for simplicity, we write $P(\boldsymbol{s}|\text{do}(\pi)) = P(\boldsymbol{s}|\text{do}(\pi); M)$. The imitating policy $\pi$ is obtainable by solving the equation $P(\boldsymbol{s}|\text{do}(\pi)) = P(\boldsymbol{s})$. We could derive a closed-form formula for $P(\boldsymbol{s}|\text{do}(\pi))$ following standard causal identification algorithms in [41, 34, 6]. As an

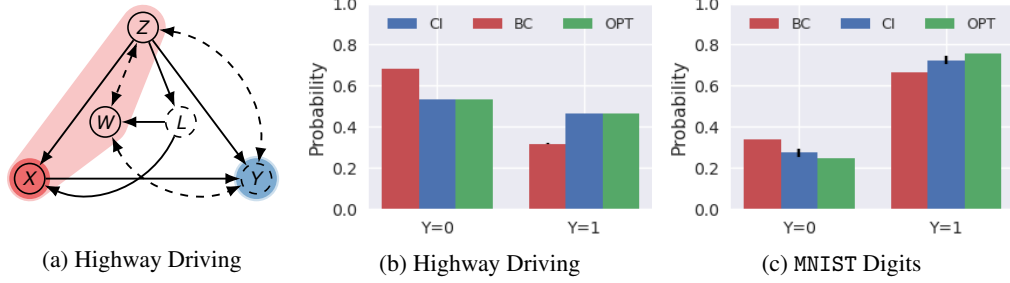

(a) Highway Driving  (b) Highway Driving  (c) MNIST Digits

Figure 4: (a) Causal diagram for highway driving example where a left-side car exists; (b,c) $P(y|\text{do}(\pi))$ induced by the causal imitation method (*ci*) and the naive behavior cloning (*bc*) compared with the actual distribution $P(y)$ over the expert's reward (*opt*).

example, consider again the setting of Fig. 1c with binary $X, W, S, Z$; parameters of $P(x, w, s, z)$ could be summarized using an 8-entry probability table. The imitating policy $\pi(x)$ is thus a solution of a series of linear equations $\sum_x \pi(x)P(s|\text{do}(x)) = P(s)$ and $\sum_x \pi(x) = 1$, given by:

$$\pi(x_0) = \frac{P(s_1) - P(s_1|\text{do}(x_0))}{P(s_1|\text{do}(x_1)) - P(s_1|\text{do}(x_0))}, \qquad \pi(x_1) = \frac{P(s_1|\text{do}(x_1)) - P(s_1)}{P(s_1|\text{do}(x_1)) - P(s_1|\text{do}(x_0))}.$$

Among quantities in the above equation, $x_i, s_j$ represent assignments $X = i, S = j$ for $i, j \in \{0, 1\}$. The interventional distribution $P(s|\text{do}(x))$ could be identified from $P(x, w, s, z)$ using the front-door adjustment formula $P(s|\text{do}(x)) = \sum_w P(w|x) \sum_{x'} P(s|x', w)P(x')$ [29, Thm. 3.3.4].

However, evaluating interventional probabilities $P(s|\text{do}(\pi))$ from the observational distribution $P(o)$ could be computational challenging if some variables in $O$ are high-dimensional (e.g., $W$ in Fig. 1d). Properties of the imitation instrument $\langle S, \Pi' \rangle$ suggest a practical approach to address this issue. Since $P(s|\text{do}(\pi))$ is identifiable w.r.t. $\langle \mathcal{G}, \Pi' \rangle$, by Def. 2, it remains invariant over the models in the hypothesis class $\mathscr{M}_{\langle \mathcal{G} \rangle}$. This means that we could compute interventional probabilities $P(s|\text{do}(\pi); \tilde{M})$ in an arbitrary model $\tilde{M} \in \mathscr{M}_{\langle \mathcal{G}, P \rangle}$; such an evaluation will always coincide with the actual, true causal effect $P(s|\text{do}(\pi))$ in the underlying model. This observation allows one to obtain an imitating policy through the direct parametrization of POSCMs [21]. Let $\mathscr{N}_{\langle \mathcal{G} \rangle}$ be a parametrized subfamily of POSCMs in $\mathscr{M}_{\langle \mathcal{G} \rangle}$. We could obtain an POSCM $\tilde{M} \in \mathscr{N}_{\langle \mathcal{G} \rangle}$ such that its observational distribution $P(o; \tilde{M}) = P(o)$; an imitating policy $\pi$ is then computed in the parametrized model $\tilde{M}$. Corol. 3 shows that such a policy $\pi$ is an imitating policy for the expert's reward $P(y)$.

**Corollary 3.** *Given a causal diagram $\mathcal{G}$, a policy space $\Pi$, and an observational distribution $P(o)$, let $\langle S, \Pi' \rangle$ be an instrument w.r.t. $\langle \mathcal{G}, \Pi \rangle$. If there exists a POSCM $M \in \mathscr{M}_{\langle \mathcal{G}, P \rangle}$ and a policy $\pi \in \Pi'$ such that $P(s|\text{do}(\pi); M) = P(s)$, then $P(y)$ is p-imitable w.r.t. $\langle \mathcal{G}, \Pi, P(o) \rangle$. Moreover, $\pi$ is an imitating policy for $P(y)$ w.r.t. $\langle \mathcal{G}, \Pi, P(o) \rangle$.*

In practical experiments, we consider a parametrized family of POSCMs $\mathscr{N}_{\langle \mathcal{G} \rangle}$ where functions associated with each observed variable in $O$ are parametrized by a family of neural networks, similar to [21]. Using the computational framework of *Generative Adversarial Networks* (GANs) [9, 27], we obtain a model $\tilde{M} \in \mathscr{N}_{\langle \mathcal{G} \rangle}$ satisfying the observational constraints $P(o; \tilde{M}) = P(o)$. The imitating policy is trained through explicit interventions in the learned model $\tilde{M}$; a different GAN is then deployed to optimize the policy $\pi$ so that it imitates the observed trajectories drawn from $P(s)$.

## 4  Experiments

We demonstrate our algorithms on several synthetic datasets, including `highD` [18] consisting of natural trajectories of human driven vehicles, and on `MNIST` digits. In all experiments, we test our causal imitation method (*ci*): we apply Thm. 2 when there exists an $\pi$-backdoor admissible set; otherwise, Alg. 1 is used to leverage the observational distribution. As a baseline, we also include naïve behavior cloning (*bc*) that mimics the observed conditional distribution $P(x|pa(\Pi))$, as well as the actual reward distribution generated by an expert (*opt*). We found that our algorithms consistently imitate distributions over the expert's reward in imitable (p-imitable) cases; and p-imitable instances commonly exist. We refer readers to [15, Appendix D] for more experiments, details, and analysis.

**Highway Driving**   We consider a modified example of the drone recordings of human-driven cars in Sec. 1 where the driver's braking action $W$ of the left-side car is also observed. Fig. 4a shows the causal diagram of this environment; $Z$ represent the velocity of the front-car; action $X$ represents the velocity of the driving car; $W$ and the reward signal $Y$ are both affected by an unobserved confounder $U$, representing the weather condition. In Fig. 4a, $\{Z\}$ is $\pi$-backdoor admissible while $\{Z, W\}$ is not due to active path $X \leftarrow L \rightarrow W \leftrightarrow Y$. We obtain policies for the causal and naive imitators training two separate GANs. Distributions $P(y|\text{do}(\pi))$ induced by all algorithms are reported in Fig. 4b. We also measure the L1 distance between $P(y|\text{do}(\pi))$ and the expert's reward $P(y)$. We find that the causal approach (*ci*), using input set $\{Z\}$, successfully imitates $P(y)$ (L1 = 0.0018). As expected, the naive approach (*bc*) utilizing all covariates $\{Z, W\}$ is unable to imitate the expert (L1 = 0.2937).

**MNIST Digits**   We consider an instance of Fig. 1c where $X, S, Y$ are binary variables; binary values of $W$ are replaced with corresponding images of `MNIST` digits (pictures of 1 or 0), determined based on the action $X$. For the causal imitator (*ci*), we learn a POSCM $\hat{M}$ such that $P(x, w, s; \hat{M}) = P(x, w, s)$. To obtain $\hat{M}$, we train a GAN to imitate the observational distribution $P(x, w, s)$, with a separate generator for each $X, W, S$. We then train a separate discriminator measuring the distance between observed trajectories $P(s)$ and interventional distribution $P(s|\text{do}(\pi); \hat{M})$ over the surrogate $\{S\}$. The imitating policy is obtained by minimizing such a distance. Distributions $P(y|\text{do}(\pi))$ induced by all algorithms are reported in Fig. 4c. We find that the causal approach (*ci*) successfully imitates $P(y)$ (L1 = 0.0634). As expected, the naive approach (*bc*) mimicking distribution $P(x)$ is unable to imitate the expert (L1 = 0.1900).

## 5   Conclusion

We investigate the imitation learning in the semantics of structural causal models. The goal is to find an imitating policy that mimics the expert behaviors from combinations of demonstration data and qualitative knowledge about the data-generating process represented as a causal diagram. We provide a graphical criterion that is complete (i.e., sufficient and necessary) for determining the feasibility of learning an imitating policy that mimics the expert's performance. We also study a data-dependent notion of imitability depending on the observational distribution. An efficient algorithm is introduced which finds an imitating policy, by exploiting quantitative knowledge contained in the observational data and the presence of surrogate endpoints. Finally, we propose a practical procedure for estimating such an imitating policy from observed trajectories of the expert's demonstrations.

## Broader Impact

This paper investigates the theoretical framework of learning a policy that imitates the distribution over a primary outcome from natural trajectories of an expert demonstrator, even when the primary outcome itself is unobserved and input covariates used by the expert determining original values of the action are unknown. Since in practice, the actual reward is often unspecified and the learner and the demonstrator rarely observe the environment in the same fashion, our methods are likely to increase the progress of automated decision systems. Such systems may be applicable to various fields, including the development of autonomous vehicle, industrial automation and the management of chronic disease. These applications may have a broad spectrum of societal implications. The adoption of autonomous driving and industrial automation systems could save cost and reduce risks such as occupational injuries; while it could also create unemployment. Treatment recommendation in the clinical decision support system could certainly alleviate the stress on the healthcare workers. However, this also raise questions concerning with the accountability in case of medical malpractice; collection of private personal information could also make the hospital database valuable targets for malicious hackers. Overall, we would encourage research to understand the risks arising from automated decision systems and mitigations for its negative impact.

Recently, there is a growing amount of dataset of natural vehicle trajectories like `highD` [18] being licensed for commercial use. An immediate positive impact of this work is that we discuss potential risk of training decision-making policy from the observational data due to the presence of unobserved confounding, as shown in Sec. 1 and 4. More broadly, since our method is based on the semantics of structural causal models [29, Ch. 7], its adoption could cultivate machine learning practitioners with proper training in causal reasoning. A favorable characteristic of causal inference methods is

that they are inherently robust: for example, the definition of imitability Def. 3 requires the imitating policy to perfectly mimics the expert performance in *any* model compatible to the causal diagram. Automated decision systems using the causal inference methods prioritize the safety and robustness in decision-making, which is increasingly essential since the use of black-box AI systems is prevalent and our understandings of their potential implications are still limited.

## Acknowledge

The authors were partially supported by grants from NSF IIS-1704352 and IIS-1750807 (CAREER).

## Footnotes

[1]Some recent progress in the field of causal imitation has been reported, albeit oblivious to the phenomenon described above and our contributions. Some work considered settings in which the input to the expert policy is fully observed [7], while another assumed that the primary outcome is observed (e.g., $Y$ in Fig. 1a) [8].

[2]This definition will facilitate the more explicitly articulation of which endogenous variables are available to the demonstrator and corresponding policy at each point in time.

[3]$\mathcal{G}_{\overline{X}}$ is a causal diagram associated with the submodel $M_x$ induced by intervention $do(x)$.

[4]The imitation is trivial if $Y \notin De(X)_{\mathcal{G}}$: by Rule 3 of [29, Thm. 3.4.1] (or [6, Thm. 1]), $P(y|\mathrm{do}(\pi)) = P(y)$ for any policy $\pi$. This paper aims to find a *specific* $\pi$ satisfying $P(y|\mathrm{do}(\pi)) = P(y)$ even when $Y \in De(X)_{\mathcal{G}}$.

[5]Consider a variation of the model where $P(U_W = 1) = 0.7$. $P(y)$ is p-imitable with $\pi(X = 0) = 0.75$.

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
