[Reviews · NeurIPS 2020]

Review 1

Summary and Contributions: This paper studied the problem of imitation learning when sensory inputs of the learner differs from those of the expert. The authors provided a non-parametric, graphical criterion that is complete for determining the feasibility of imitation learning on the combinations of demonstration data and the assumptions about the underlying environment. When the criterion does not hold, the authors further proved that imitation could still be possible by exploiting quantitative knowledge of the expert trajectories.

Strengths: This paper provided some theoretical results on how to judge whether or not imitation learning on the combinations of demonstration data and the assumptions about the underlying causal models of the environment is feasible under different definitions of imitability. Accordingly, they also provided an algorithm to learn the imitation policy based on the theoretical results.

Weaknesses: Generally speaking, I have several doubts in the current paper. 1. In the paper the authors explored the problem of imitation learning, based on a very restrictive assumption that the causal diagram is given. This is highly impossible in most real world applications, which renders the proposed approach not that useful on the practical side. 2. Even if we assume that we know the causal models on the observed variables, how could we know how many hidden confounders exist among them and how hidden confounders influence among them? Apparently, hidden confounders play an extremely essential role in the proposed method in this paper. Without such knowledge, it is impossible to accurately find the i-backdoor admissible set and the imitation surrogate set. 3. All the examples (Figs 1a, 1b, 1c, 1d, 3a, 3b, 4a) presented in the paper appear hard to find their corresponding cases in the real world, in particular for those examples in which there exist an imitation surrogate or instrument for the outcome and in which actions did not rely on anything. 4. The three points above immediately bring us a more important question in the setting of this paper: what if the assumed causal graph did not match the true causal graph of the task? I think this question is too important for practical applicability to ignore in the current setting. The authors should take them into consideration, e.g., providing some error bounds for the learned policy in the case of mismatched causal graph. 5. Considering that the proposed approach is eventually reduced to searching the imitation surrogate or instrument among observed variables, in some sense it is kind of the problem of causal feature selection. There are already several approaches proposed to deal with this problem, e.g., the one in the causal confusion in imitation, the one in ICP, and the one in causal invariant prediction in block mdps or the one in IRM, etc. It is necessary to compare with them. 6. I am curious that in the experiments the authors did not compare with any SOTA approach to imitation learning like GAIL. Also they did not conduct any experiments on the popular RL benchmarks and environments for imitation learning, e.g., OpenAI gym, MuJoCo, and Atari etc. Without any comparison like these, we are unable to judge the practical implications of the proposed approach in the applications.

Correctness: Most of them seem sound. 1. The description on line 56 does not match Figure 1a, where L depends only on Z in the figure whist L corresponds to a function of Z and U. This mismatch makes the claim not that convincing. 2. On line 168, the claim that P(Y=1|do(\pin))=0.5 for any policy \pi is inaccurate. This claim is only correct under the assumption that \pi is defined as P(x) without any conditional. Apparently, when \pi is defined as P(x|u) where u is the common cause of X and Y, we can directly derive that P(Y|do(\pi))=P(Y). There might exist some other scenarios (e.g., instrument variables) in which the claim is wrong. Here, this claim should be clarified up. 3. It seems that for Fig 3b, in both settings (line 167 and line 225-226) observational probabilities P(x,w,y) are provided. So what do you mean by "new imitation opportunities arise when observational probabilities P(x, w, y) are provide" on line 225-226? Why are their claims so different? Also, the setting on line 225-226 does not match the graph in Fig 3b, I therefore did not see any connection.

Clarity: The paper is not friendly to readers unfamiliar with causality, in particular do-calculus, as many claims presented in the paper left out the derivation process.

Relation to Prior Work: The authors should add more details on the traditional approaches to imitation learning, in particular on feature selection of imitation learning.

Reproducibility: No

Additional Feedback:


Review 2

Summary and Contributions: This paper proposes an imitation learning strategy for settings where the demonstrator is able to observe state variables that the imitator cannot. A naive graphical criterion for "imitability" will only be satisfied if the causal graph (which is assumed to be supplied) admits an "i-backdoor" between the demonstrator's action and latent reward (Thm. 2). However, an imitation policy can still be obtained if one, (a) weakens the notion of imitability to be specific to one particular observational distribution (Defn. 5); (b) is able to find a "surrogate" set of observable variables that decouple the demonstrator's action from the latent reward; and (c) constrains the policy space to ensure that the distribution over surrogates (after intervening on pi) is identifiable. These principles, combined with two observations that constrain the search space for the surrogate set and policy subspace (Lem. 3, Lem. 4), lead to a practical algorithm that is validated on synthetic and semi-synthetic data.

Strengths: Provides a thorough exploration of the graphical conditions (and distribution constraints) enable "optimal" imitation, in the sense that P(y | do(pi)) = P(y). Theorem 5 in particular provides a non-trivial constraint on what future algorithms can achieve using only knowledge of the causal graph (or causal discovery methods that obtain a causal graph). - The proposed algorithm could be useful in settings where human demonstrators are able to observe facets of the world that robots cannot. This problem ought to be reasonably common given the limitations of robotic sensors, and there is already interest in this setting in the imitation learning community (search "third-person imitation").

Weaknesses: From an imitation learning perspective, the notions of imitability and p-imitability are excessively strong. In practice, one is typically not interested in ensuring that P(y | do(pi)) = P(y), but rather in maximising E[y | pi]. The relatively strong conditions required for Algorithm 1 (existence of an i-instrument, availability of the causal graph) will still rule out many situations where "imitation" (in the colloquial sense of successfully solving roughly the same task as the demonstrator) is still possible. Relatedly, the requirement for a causal graph that _includes the latent reward variable_ is also very strong. Imitation learning is useful in settings where it's hard to define an appropriate reward function by hand, and therefore hard to do planning/RL directly. However, this method additionally requires that it's easy to identify a small subset of state variables that the reward function could depend on, which seems like a setting where it would also be easy to define the reward function by hand. - The specific motivating example used in the introduction is dubious—see comments on related work. - The writing is difficult to follow at times—see comments on clarity.

Correctness: The theoretical and algorithmic contributions appear sound to me, with the caveat that I did not closely check the proofs in the appendix.

Clarity: The key ideas are discernable, but the paper is at times difficult to follow. Some examples: - Some of the wording is circuitous. For instance, line 30: "An alternative approach used to circumvent this issue is to find a policy that mimics a demonstrator’s behavior, which leads to the imitation learning paradigm [2, 4, 12, 23]; that is, how one should undertake imitation." This could be reduced to "This issue may be overcome using *imitation learning*, which finds a policy that mimics the demonstrator's actions." Ambiguous order of quantifiers is a related problem. For example, does L152-–154 mean "for all M in M(G), pi should mimic the expert reward P(y; M)" or "pi should be chosen such that for all M in M(G), pi mimics expert reward P(y; M)"? - Variable names: for readers coming from an imitation learning or robotics background, it would be helpful to use variable names that match the conventions of those fields. For instance, $U$ or $A$ for control/action, $X$ or $S$ for observable state, and $R$, $U$ or $C$ for reward/utility/cost (some of these choices would conflict with conventions like using $U$ for exogeneous variables, but I'm sure a reasonable combination can be found). - The paper has been heavily compressed to fit within the page limit, often at the expense of readability (e.g. "for $i=1,2$" on L161, and long inline expressions like on L142). This could be addressed by being slightly more aggressive in moving material to the appendix (which is still not ideal, given that the appendix is already 17 pages), or by publishing in a venue like JMLR that provides a more appropriate amount of space. While these factors do not affect the technical merit of the paper, they did make it much harder to review, and will presumably make it harder for future readers to understand.

Relation to Prior Work: I'm not aware of directly related work beyond [6] and [7], which are mentioned in a footnote in the introduction. The difference from de Haan et al. [6] is clear—they seem to be doing what is suggested by Theorem 1, with some variable selection tricks to remove weak (spurious) correlates. On the other hand, the paper would be more convincing if there was some justification of the differences between this method and that of Etesami and Geiger [7]. Regarding [7]: my understanding is that this paper assumes that the "demonstration observer" (the drone, in the case of HighD) can observe the same variables as the imitator (an autonomous vehicle, say), but the demonstrator (a human-driven vehicle) might be able to observe variables that neither the demonstration observer nor the imitator can observe. In contrast, [7] focuses on the setting where the demonstrator and the imitator can observe the same variables when acting, but the "demonstration observer" cannot observe some variables. The setting of [7] seems like a more natural fit for the HighD example used to motivate the approach in the intro of this paper: an AV will be able to use cameras to observe roughly the same variables as a human driver, but it might still be desirable to leverage training data that was recorded from a different perspective, such as a drone or highway overpass camera. I expect there are still settings where the approach proposed in this paper is useful, but the introduction could be improved to make this more clearly evident.

Reproducibility: Yes

Additional Feedback: I believe the accessibility of the paper to an imitation learning audience could be improved by addressing the concerns raised above regarding clarity and motivation. I have read the response from the authors. My decision remains unchanged.


Review 3

Summary and Contributions: Success of imitation learning relies on expert demonstrations. It is often assumed that the observed covariates for learner and expert are the same. This paper explores the scenario when the above is not true and when there are unobserved confounders in the environment. This work provides a theoretical framework for causal imitation learning by defining a partially observable structural causal model (POSCM) of the environment. The assumptions of the model and the criterion for determining the feasibility of learning an optimal imitating policy in the presence of a causal model is clearly stated. The paper also presents an algorithm for learning an imitating policy when such criterion does not hold. By applying causal inference methods, the proposed framework is expected to be robust than existing algorithms.

Strengths: The framework of POSCM is novel and sufficient proofs have been provided to the claims made in the paper. The theorems are well defined and have been used to support the algorithm of learning an optimal imitating policy from a causal model This work addresses the problem of unobserved confounders that leads to learning a suboptimal policy in imitation learning. The framework can be adopted into building autonomous systems and clinical decision support Performance of the algorithm on sample datasets were evaluated and compared to naïve imitation learners and is in par with optimal performance The work has also discussed the subsequent societal impacts of using such algorithms in industrial systems

Weaknesses: Scalability of the solution to real time applications is challenging . Any limitations or challenges of the proposed approach are not very well discussed in the work.

Correctness: Yes. The methodology closely follows the approach used in current causal inference literature.

Clarity: The paper is well written with sufficient examples and illustrations as required. The motivation for the work is quite clear. The sentence structure could have been simplified.

Relation to Prior Work: There is a comparison to traditional imitation learning methods such as behavior cloning. The previous contributions or attempts in causal imitation learning is not very well discussed.

Reproducibility: Yes

Additional Feedback:


Review 4

Summary and Contributions: The paper provides a framework to evaluate imitation learning problems for feasibility of imitation depending on the observable covariates, using tools from causal inference In particular, they highlight properties that make problems identifiable or imitatible.

Strengths: The paper nicely ties features of causal learning and imitation learning, too fields that have a large audience at Neurips. The framework seems relatively novel and would be a good tool for analysis. The framework provided by the paper allows one to analyze an imitation learning problem for feasibility based on the observable features. Imitation learning is a core technique for learning policies in the real-world and issues with causal identification are central. While additional contributions of the paper might be limited, this seems like an interesting area to explore further.

Weaknesses: While the work provides an interesting framework in which to study and analyze imitation learning problems, I imagine the practical applications are limited. I found the algorithm section of section 3.1 / algorithm 1 relatively confusing, and from the given implementations in the experiments this seems to result in a GAN algorithm that does state / observation matching, quite similar to GAIL in some respects which is not mentioned.

Correctness: The theoretical claims seem correct, and the proofs relatively direct. However, causal inference is not a specialty of mine and probably others may be better qualified to review in more detail.

Clarity: The paper seems well written, the intuitive examples are useful though there could be more. The paper, while dense, does seem to be nicely explained and broken down into important contributions

Relation to Prior Work: I think some more discussion on comparing with imitation learning algorithms is warranted. For example, while this does allow for analysis of the problems, the underlying algorithm seems like it overlaps with existing imitation algorithms

Reproducibility: Yes

Additional Feedback:

[Author Response · NeurIPS 2020]

We thank the reviewers for their thoughtful feedback. We believe that a few misreadings of our work made some of evaluations overly harsh and would ask reviewers to reconsider our paper in the light of clarifications provided below.

**R1:** We organize the reviewer's questions (Q) and provide answers below. Point #6 clarifies questions in "Correctness".

**1. Are graphs necessary? (Q1-2, Q4)** The departing point of our work is the realization that an imitating policy is generally underdetermined by the observational data alone. For concreteness, consider models $M_1, M_2$, unknown to researchers, where in $M_1$, $X \leftarrow U$, $Y \leftarrow X$; in $M_2$, $X \leftarrow U$, $Y \leftarrow X \oplus U$; in $M_i, i = 1, 2$, $P(U = 0) = P(U = 1) = 0.5$. We assume that $Y, U$ are unobserved; $Y$ is the reward. In both $M_1$ and $M_2$, the observational distribution $P(X = 0) = P(X = 1) = 0.5$. In $M_1$, $P(y)$ is imitable with policy $\pi(x) = P(x)$; while in $M_2$, $P(Y = 0|\text{do}(\pi)) = 0.5$ for any $\pi(x)$, which is far from $P(Y = 0) = 1$. This example shows that when unobserved confounders (UCs) are present, imitation learning from observations alone is generally impossible.

**2. Learning causal diagrams (Q1-2, Q4)** A common approach to address the challenges of UCs is to explore functional relationships among variables, represented as a causal diagram. For instance, $M_1, M_2$ in the previous example would be distinguished using their corresponding causal diagrams $\mathcal{G}_1, \mathcal{G}_2$ (since bi-directed arrow $X \leftrightarrow Y$ exists in $\mathcal{G}_2$ but not $\mathcal{G}_1$). One could construct causal diagrams with the assistance of domain experts; see examples in (Bottou et al., 2013). In addition, efficient methods for learning causal diagrams from data have been studied under the rubrics of causal discovery (Spirtes, Glymour, and Scheines, 2000), so that dependence on domain experts is minimized. Our methods could be combined with causal discovery algorithms to learn an imitating policy after the graph is obtained. Finally, to address the challenges of incorrect causal diagrams, one could apply causal discovery methods to detect "model misspecifications", i.e., incompatibilities between the diagram and the collected data.

**3. Examples, benchmarks (Q3, Q6)** Since most RL benchmarks do not explicitly model the presence of UCs, we study the highD dataset which contains trajectories of human driving. We demonstrate how challenges of UCs could arise when applying imitation learning with highD, some of which have been acknowledged in (Etesami and Geiger, 2020). Overall, we agree that imitation learning from observational data contaminated with UCs in sequential decision-making settings is an important and pervasive challenge. This paper takes the first step towards a solution by studying the feasibility of learning an imitating policy from observational data when the causal diagram is obtained.

**4. GAIL (Q3, Q6)** In all experiments (Sec. 4 and Appendix D), the learner $bc$ is able to learn the nominal expert's policy $P(x|pa(\Pi))$, but still diverges significantly in the performance $P(y)$. Since GAIL is not concerned with UCs, it converges to the nominal policy $P(x|pa(\Pi))$, which means that its performance coincides with $bc$. Having said that, our methods could certainly be combined with GAIL to ensure both the causal robustness and the scalability with high-dimensional data, which we'll acknowledge in the paper. In particular, once an i-backdoor/i-strument is found, one could then apply GAIL to obtain a policy that imitate the expert's performance.

**5. Related work (Q5)** We appreciate the suggested references, but they are somewhat orthogonal to our problem. As we mentioned in Footnote 1, (de Haan, Jayaraman, and Levine, 2019) assume that "input to the expert policy is fully observed", i.e., there is no unobserved confounding. (Arjovsky et al., 2019; Zhang et al., 2020) attempt to find a set of variables so that adjustment on them leads to invariant parameters across multiple environments. Non-parametric methods for such problem have been studied under the rubrics of transportability (Pearl and Bareinboim, 2011).

**6. Other comments** (1) $U$ is an independent exogenous noise affecting only $L$, which "are not explicitly shown" in a causal diagram (Line 110-112) (2) By "any policy", we mean a policy taking observed variables (i.e., $Z$) as input. (3) A causal diagram is imitable where an imitating policy exists regardless of parameters of the observational distribution (e.g., $P(x, w, y)$). Fig. 3(b) is not imitable due to the counterexample in Line 167. However, one could still find an imitating policy in Fig. 3(b) for some *specific* $P(x, w, y)$ (called p-imitable); Line 255-256 shows such an example.

**R2:** (1) A causal diagram containing latent reward $Y$ generalize the traditional settings of imitation learning. For instance, inverse reinforcement learning requires the parametric form of the reward function, and all input must be observed. (2) While it is not complete, Alg. 1 explores settings where existing imitation methods are not applicable, i.e., when unobserved confounding is present. (2) (Etesami and Geiger, 2020) assumes that reward $Y$ is observed, and the goal is to ensure $P(y) = P(y|\text{do}(\pi))$. This paper assumes $Y$ is unobserved and develop non-trivial methods to find its surrogates (Def. 6). Also, (Etesami and Geiger, 2020) consider a canonical causal diagram and explore parametric assumptions (e.g., linearity); while we focus on non-parametric causal identification methods in an arbitrary graph.

**R3:** We appreciate your feedback and suggestion. A possible solution to scalability is discussed in #4 to R1. Thanks.

**R4:** We respectfully disagree with the statement that "practical applications" of our methods "are limited." As we discussed in #2 (to R1), there are causal discovery efficient algorithms to construct causal diagrams from the data; methods could be applied after a graph is obtained. Nevertheless, UCs are still present in the environment despite the hardness of finding its causal diagram, as demonstrated in #1. This paper explicitly acknowledges the existence of UCs and provides methods to circumvent its possibly nefarious effects. As for a discussion on GAIL, please refer to #4.

[Meta-Review · NeurIPS 2020]

summary: This paper studies the feasibility of imitation learning for decision making from a causal perspective. The paper consideras a very general setting with possible unobserved confounders, expert and policy can have different inputs and the reward being unobserved. The work presents multiple criteria for ensuring successful imitation in particular based on proxy variables for task rewards. meta review: I think this is a very important paper, and I want to recommend it for an oral presentation for the following reasons in spite of the review scores of 5, 7, 7, 6: - Imitation learning is an essential method in reinforcement learning, used eg when expert demonstrations are available but the utility function is difficult to specify explicitly (often the case for complex tasks in robotics, or in a multi-agent / social learning context) or even as sub-routines of other RL approaches (getting learning faster off the ground [Mastering the game of Go with deep neural networks and tree search. Nature] or in kick-starting hard exploration problems). - Imitation learning can fail and characterization of failure cases have not been studied prior; furthermore, AFAIK there is little awareness in the applied RL community that imitation learning is not guaranteed to work. This work can raise awareness of this fundamental problem. - The paper presents multiple, clearly formulated sufficient conditions for imitation learning to succeed. The criteria represent a large advancement in our understanding how to identify situations / assumptions sufficient for imitation learning. Therefore, I expect this paper to have a large impact in the NeurIPS community. All reviewers (apart from Rev 1, reasons for exclusion below) agree that the paper addresses an important topic and that it gives valuable insights into possible solutions. The main reservations voiced by Rev 1 and somewhat by Rev 3&4 that “assumptions are too strong”, “practical application” could be limited and that there could be issues with “limited scalability” have to compared to current practice of imitation learning as applied in RL, where its feasibility (or necessary assumptions) is not studied at all, replaced by a hope that it’ll just work. The authors convincingly show with a small example that without any assumptions, imitation learning can fail, and the only way to make progress is to formulate sensible assumptions. I therefore recommend to discard this criticism of the work and thereby to heavily discount the score from Rev 1.